# Patient-Perceived Impact of the COVID-19 Pandemic on Medication Adherence and Access to Care for Long-Term Diseases: A Cross-Sectional Online Survey

**Beatriz Santos** [1,2,3], **Younes Boulaguiem** [4], **Helene Baysson** [5], **Nick Pullen** [5], **Idris Guessous** [6,7], **Stephane Guerrier** [2,4], **Silvia Stringhini** [5,7,8] **and Marie P. Schneider** [1,2,3,*]

1 School of Pharmaceutical Sciences, University of Geneva, 1205 Geneva, Switzerland; beatriz.santos@unige.ch
2 Institute of Pharmaceutical Sciences of Western Switzerland, University of Geneva, 1205 Geneva, Switzerland; stephane.guerrier@unige.ch
3 Pharma24, Academic Community Pharmacy, 1205 Geneva, Switzerland
4 Geneva School of Economics and Management, University of Geneva, 40 Boulevard du Pont d'Arve, 1204 Geneva, Switzerland; younes.boulaguiem@unige.ch
5 Unit of Population Epidemiology, Division and Department of Primary Care Medicine, Geneva University Hospitals,1205 Geneva, Switzerland; helenemariepierre.baysson@hug.ch (H.B.); silvia.stringhini@unige.ch (S.S.)
6 Division of Primary Care Medicine, Geneva University Hospitals, 1205 Geneva, Switzerland
7 Department of Health and Community Medicine, Faculty of Medicine, University of Geneva, 1205 Geneva, Switzerland
8 School of Population and Public Health and Edwin S.H. Leong Centre for Healthy Aging, Faculty of Medicine, University of British Columbia, Vancouver, BC V6T 1Z4, Canada
* Correspondence: marie.schneider@unige.ch

**Abstract:** The COVID-19 pandemic has been associated with lifestyle changes, reduced access to care and potential impacts on medication self-management. Our main objectives are to evaluate the impact of the pandemic on patient adherence and access to care and long-term medications and determine its association with sociodemographic and clinical factors. This study is part of the Specchio-COVID-19 longitudinal cohort study in Geneva, Switzerland, conducted through an online questionnaire. Among the 982 participants (median age: 56; 61% female), 827 took long-term medications. There were 76 reported changes in medication dosages, of which 24 (31%) were without a physician's recommendation, and 51 delays in initiation or premature medication interruptions, of which 24 (47%) were without a physician's recommendation. Only 1% (9/827) of participants faced medication access issues. Participants taking a respiratory medication had a four-times greater odds of reporting more regular medication (OR = 4.27; CI 95%: 2.11–8.63) intake, whereas each year increase in age was significantly associated with 6% fewer relative risks of discontinuation (OR = 0.94; CI 95%: 0.91–0.97) and 3% fewer relative risks of changes in medication dosage (OR = 0.97; CI 95%: 0.95–1.00). Despite the limited impact of the pandemic on adherence and access to medications, our results emphasize the need for understanding patient challenges when self-managing their long-term medication, notably during public health crises.

**Keywords:** COVID-19; medication adherence; lockdown; impact; outpatients; access to care

## 1. Introduction

Nonadherence is a silent epidemic, with approximately 40 to 50% of people with long-term diseases being not optimally adherent to their medications. Yet, medication adherence is a key component of long-term disease management [1].

The impact of the COVID-19 pandemic on medication adherence and access to care is a crucial area of study given its implications for patient care. Several factors have been identified as barriers to healthcare and medication management during the pandemic, such

as access to healthcare providers, medication supplies and travel restrictions [2,3]. However, little is still known about the potential impact on adherence and access to care for individuals with long-term conditions. Research shows significant differences in the impact of the pandemic on medication adherence by type of medication, care setting and geographical context. A systematic review evaluated twelve studies from different countries, of which seven reported a worsening in adherence in the pandemic period compared with previous years, while five studies showed no change or an improvement in adherence during the COVID-19 pandemic [4]. While most studies have focused on specific conditions, such as anxiety and depression [5] and inflammatory diseases [6,7], or on certain types of medications, such as immunosuppressive and biological therapies [8–10], there is still limited research on the impact of the pandemic on the general chronic outpatient population.

With major differences between healthcare systems in Europe alone, as well as differences in the containment measures implemented, it is difficult to generalize the impact of the COVID-19 pandemic on medication adherence in the outpatient population. In Switzerland, the Federal Council implemented a partial lockdown in March 2020 to prevent the spread of SARS-CoV-2, which consisted of a strong recommendation to stay at home and a series of containment measures such as physical distance, masks, self-isolation or travel restrictions. These lockdown measures have been associated with social isolation, often resulting in psychological distress and deep lifestyle changes. In May 2020, the World Health Organization (WHO) conducted a survey of 155 countries on access to noncommunicable disease (NCD) services and found that COVID-19 had a marked impact on these services across all regions and income groups [11]. In three-quarters of these countries, NCD services (e.g., rehabilitation services, hypertension management, diabetes care, asthma services) were significantly affected. The lockdown also limited access to essential care and decreased medication adherence, whether by discontinuing medications because of a lack of follow-up or by not being able to refill the prescription [12,13]. Furthermore, patients were confronted with misinformation about the COVID-19 pandemic, which could have changed their motivation to continue taking their long-term medication as prescribed [14].

This study aims to provide a more comprehensive understanding of the impact of the pandemic on medication adherence by examining several factors from an outpatient perspective. This can help identify areas where patients may need additional support to maintain medication adherence during a health crisis such as a pandemic. Therefore, the objective of this study, conducted in the canton of Geneva, Switzerland, was to explore and evaluate the impact of the COVID-19 pandemic and its associated lockdown measures on the following: 1. Patient adherence to long-term medications and motivation to take medications. 2. Patients' access to medications and access to medical appointments/exams and overall medical care. 3. The use of home care, therapeutic physical activity and diet support and potential associated sociodemographic and clinical factors.

## 2. Materials and Methods

### 2.1. Study Design and Patients' Eligibility

This is a cross-sectional online survey. Our population is part of the Specchio-COVID-19 serosurvey study, a population-based digital study [15] for the short- and long-term monitoring of the COVID-19 pandemic among the general population of the Canton of Geneva (around 5,000,000 inhabitants) [15]. Adult serosurvey participants were invited to take part in the Specchio-COVID-19 study after a baseline serological test. The participation was carried out through the Specchio-hub digital platform "www.specchio-hub.ch" (accessed on 6 February 2024), a website specifically created for this purpose. After registration, questionnaires are directly filled in online, where all requirements to guarantee the secured management and storage of sensitive data are met. When enrolled, participants are asked to complete a mandatory questionnaire on sociodemographic and clinical data (inclusion questionnaire). They are also invited, on a regular basis, to complete follow-up questionnaires on health and mental behaviors, COVID-19 tests and vaccination status [16]. At the

time of our study, there were around 8000 participants in the Specchio-COVID-19 cohort. The population eligible for our survey was Specchio-COVID-19 adult participants, who reported at least one long-term disease in the inclusion questionnaire (n = 2788). There were no exclusion criteria. No sample size calculations were performed as the prevalence of non-adherence was unknown, and the survey was sent to all eligible participants to recruit as many as possible for three months based on previous experience with Specchio-COVID-19 online surveys.

## 2.2. Study Variables and Measurements

We developed a 13-item (with respective subitems) multiple-choice and Likert scale self-report questionnaire (Supplementary Materials, Questionnaire Participants with Chronic Illnesses). The questionnaire is available as Supplementary Material. The survey was pre-tested by four patients to ensure clarity and understanding of the questions. In this study, we defined three components of nonadherence to medication, using the ABC taxonomy [17]: delay in medication initiation, medication premature discontinuation and unplanned self-adjustments to the treatment such as increases or decreases in medication dosage. Primary measured items were 1. delays at medication initiation, 2. irregular medication intake, 3. self-adjustments of the dosage, 4. medication discontinuation and 5. changes in motivation to take medications. Secondary measured items were 6. access to overall medical care, 7. access to medications, 8. access to medical appointments, 9. accessibility of medical exams, 10. access to home care, 11. motivation to follow the recommended diet, 12. motivation to follow physical activity programs and 13. participation in discussion support groups. As co-variables, participants were asked about their long-term diseases; medications taken; their current health status regarding mobility, autonomy, everyday activities, pain/discomfort and anxiety/depression; and their perceived risk of COVID-19 infection related to their long-term disease. Sociodemographic variables were extracted from the Specchio-COVID-19 inclusion questionnaire: (i) age; (ii) gender; (iii) nationality; (iv) native language; (v) level of education; (vi) household composition; (vii) work situation; and (viii) income and financial situation. Participants' vaccination status (0, 1 or 2 doses) and perception of their own health (general and mental health, assessed through a 5-point Likert scale) were extracted from the general health questionnaire completed between June and September 2021.

## 2.3. Data Management and Statistical Analysis

Diseases were categorized according to International Classification of Diseases (ICD-11) codes [18]. Medications were coded using the Anatomical Therapeutic Chemical Classification system (ATC-codes) [19]. We conducted descriptive analysis for sociodemographic and clinical data (mean and standard deviation and median and interquartile range, as appropriate). We used a logistic regression approach to examine the association between the different outcomes and the following variables selected according to their contextual and clinical relevance based on the existing literature and previous experiences in the Swiss healthcare system [20–22]: gender (male vs. female), age (continuous), level of education (primary and secondary vs. tertiary), living situation (alone or single parent with children <18 years old vs. with one or more adults), general health (poor or very poor general health vs. average to very good), vaccination status (vaccinated with at least one dose vs. non-vaccinated), the number of long-term medications (1 to 3 vs. more than 3), taking one or more medications for the respiratory system and reporting financial difficulties. Variables such as gender, age, education level and financial status were selected because their association with patient behavior has been described in several long-term conditions [1,23–25]. COVID-19 vaccination status and use of respiratory medications were also selected because of their association with SARS-CoV-2 symptom severity and potentially with patient adherence behavior [26,27]. To ensure the accuracy and validity of the regressions, an EPV (number of events per variable) score of 5 was applied, which means that a minimum number of 5 events (outcomes) per predictor variable was used in the logistic regression model [28]. This limited the choice of independent variables in

the logistic model. The absence of answers and answers of "I don't know" were labeled as missing data and removed from the analysis. The statistical analysis was performed in R (version 4.0.2) using binary (if the dependent variable had two levels) and multinomial (for more than two levels of dependent variables) logistic regressions. A variable with a *p*-value of less than 0.05 was considered statistically significant. We use the generalized Holsmer–Lemeshow goodness-of-fit test for logistic regression (binary and multinomial) using the generalhoslem package in R (version 4.0.2). A *p*-value greater than the significance level suggests that the logistic regression model's fit is consistent with the observed data.

## 3. Results

The online questionnaire was sent to participants in mid-September 2021 with a reminder after 7 days and was available in the participants' personal spaces for a period of 3 months. The questionnaire was sent to 2788 participants; 1549 (55%) completed the questionnaire, and 1048 confirmed having at least one long-term condition. Our baseline population consisted of 982 individuals suffering from at least one long-term condition insured in Switzerland (therefore attending healthcare in Switzerland), of whom 827 respondents were taking at least one long-term medication on a regular basis. Figure 1 presents a flowchart of the participants' inclusion.

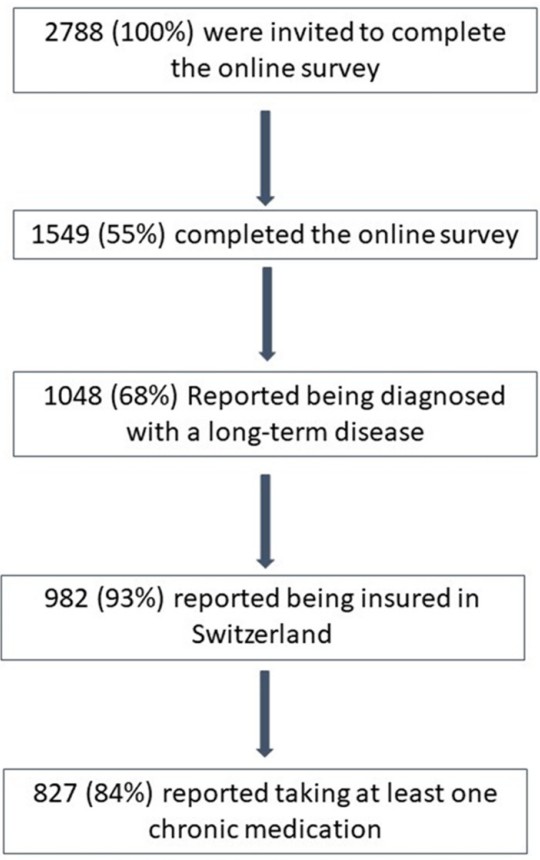

**Figure 1.** Flowchart of the inclusion of participants.

Participants had a median age of 56 years old (IQR: 47.0–66.0), and 61% were female. In total, 95% were of Swiss or European nationality, and 71% reported French as their native language. In total, 51% had a tertiary level of education (university/professional college), and 27% lived alone or as a single parent. In total, 62% of participants were either employed or in training, and 9% reported experiencing financial difficulties. Table 1 presents the sociodemographic data of the participants.

**Table 1.** Sociodemographic data.

| Sociodemographic Data | N (%) |
|---|---|
| Age range (yrs) (n = 982; min 18; max 97; missing data = 0;) | |
| 18–40 | 127 (13%) |
| 41–60 | 482 (49%) |
| 61–80 | 356 (36%) |
| ≥ 81 | 17 (2%) |
| Gender (n= 982; other = 4; missing data = 0) | |
| Female | 596 (61%) |
| Nationality (n = 982; missing data = 0) | |
| Swiss | 831 (85%) |
| European | 101 (10%) |
| Extra-European | 50 (5%) |
| Native language (n = 982; missing data = 0) | |
| French | 697 (71%) |
| Highest education level (n = 977; missing data = 5) | |
| Primary (obligatory schooling) | 41 (4%) |
| Secondary (middle school, high school, apprenticeship) | 432 (44%) |
| Tertiary (university, professional college) | 504 (51%) |
| Living as (n= 982; missing data= 0) | |
| Couple, no children | 341 (35%) |
| Couple with children | 333 (34%) |
| Alone | 210 (21%) |
| Together with other adults (cohabitation) | 46 (5%) |
| Single parent | 52 (5%) |
| Working situation (n = 982; missing data = 0) | |
| Employed | 528 (54%) |
| Retired | 286 (29%) |
| Independent | 70 (7%) |
| Disabled (disability insurance) | 25 (3%) |
| In the household | 34 (3%) |
| Unemployed | 25 (3%) |
| Student | 14 (1%) |
| Financial difficulties (n = 982; missing data = 0) | |
| No (never) | 534 (54%) |
| Yes, in the past (not anymore) | 300 (31%) |
| Yes, I have been for many years | 55 (6%) |
| Yes, for a year or less | 30 (3%) |
| I do not wish to answer | 63 (6%) |

The 982 participants mentioned a total of 2548 diseases (an average of 2.6 diseases per participant). The main long-term disease was cardiovascular disease (19%), followed by metabolic disease (15%) and osteoarticular diseases (13%). In 44% of cases, participants had been diagnosed for over 10 years. Among the 84% (n = 827/982) of participants who reported taking at least one long-term medication, 23% (n = 191) took four or more medications, mainly for the cardiovascular system (24%); the alimentary tract and metabolism (14%); and the nervous system (12%). Among individuals who reported feeling more at risk of COVID-19 infection because of their long-term conditions (n = 103/982), 14 had not been vaccinated at the time of the questionnaire. A total of 149 participants (15%) reported not

being vaccinated. General health was perceived as poor or very poor by 4% (n = 40/982) of participants, and 3% (n = 28/982) described their mental health as poor or very poor. Table 2 presents the clinical data.

**Table 2.** Clinical and health-related data.

| Clinical and Health-Related Data | N (%) |
|---|---|
| Chronic diseases (n = 2548; missing data = 0) | |
| Heart diseases | 483 (19%) |
| Metabolic diseases | 367 (14%) |
| Osteoarticular diseases | 328 (13%) |
| Respiratory diseases | 247 (10%) |
| Digestive diseases | 173 (7%) |
| Psychological/mental diseases | 172 (7%) |
| Neurological diseases | 148 (6%) |
| Urinary tract diseases | 126 (5%) |
| Dermatological diseases | 121 (5%) |
| Immunological diseases | 109 (4%) |
| Cancers | 97 (4%) |
| Others (eyes, infectious diseases, blood and others non-classified) | 177 (7%) |
| Number of medications taken (n = 917; missing data = 65) | |
| 0 | 90 (9%) |
| 1 | 317 (32%) |
| 2 | 184 (19%) |
| 3 | 135 (14%) |
| >3 | 191 (19%) |
| Types of chronic medications (n = 1878; missing data/unclassified = 23) | |
| Cardiovascular system | 538 (29%) |
| Alimentary tract and metabolism | 316 (17%) |
| Nervous system | 270 (14%) |
| Respiratory system | 154 (8%) |
| Blood and blood-forming organs | 138 (7%) |
| Musculoskeletal system | 117 (6%) |
| Systemic hormonal preparations (sex hormones and insulin excluded) | 114 (6%) |
| Others (antineoplastic/immunomodulating agents; genito-urinary system and sex hormones; anti-infectives and antiparasitic agents for systemic use; sensory organs; dermatologicals; various) | 208 (11%) |
| COVID-19 vaccination status (n = 891; missing data = 91) | |
| Two doses | 627 (64%) |
| One dose | 115 (12%) |
| None | 149 (15%) |
| Perceived risk of COVID-19 infection due to own chronic diseases (n = 981, missing data = 1) | |
| Yes, I am at risk | 103 (10%) |
| No, I am not at risk | 315 (32%) |
| No, I am not at risk because I am vaccinated | 519 (53%) |
| I do not know | 44 (4%) |
| Perceived general health (n = 982; missing data = 0) | |
| Very good | 124 (13%) |
| Good | 551 (56%) |

**Table 2.** *Cont.*

| Clinical and Health-Related Data | N (%) |
|---|---|
| Average | 267 (27%) |
| Poor | 38 (4%) |
| Very poor | 2 (0%) |
| Perceived mental health (n = 982; missing data = 0) | |
| Very good | 242 (25%) |
| Good | 528 (54%) |
| Average | 184 (19%) |
| Poor | 25 (3%) |
| Very poor | 3 (0%) |
| Participants currently reporting . . . (moderate to incapacitating) (n = 982, missing data = 0) | |
| Pain/discomfort | 234 (24%) |
| Anxiety/depression | 143 (15%) |
| Performance of current activities | 85 (9%) |
| Issues with mobility | 71 (7%) |
| Issues with autonomy | 10 (1%) |

*3.1. Impact of the Pandemic on Adherence to Long-Term Medications and Motivation to Take Medications*

Among participants, 5% (n = 38/827) reported taking their medications more regularly, while 2% (n = 20/827) reported taking their medications less regularly than before the pandemic. As presented in Table 3, participants who took medication for the respiratory system had greater odds of reporting more regular medication intake (OR = 4.27; CI 95%: 2.11–8.63), whereas older participants (each year increase in age) had lower odds of reporting less regular medication intake (OR = 0.93; CI 95%: 0.89–0.96). Additionally, 11% (n = 93) of participants reported changes in medication dosage, with 49 increases and 27 decreases (missing data = 17), of which 20% (n = 10/49) and 52% (n = 14/27), respectively, were not recommended by a physician. There were 25 reported delays in treatment initiation, and 26 reported treatment discontinuations, of which 40% (n = 10/25) and 54% (n = 14/26), respectively, were not recommended by a physician. Each year increase in age is significantly associated with less discontinuation (OR = 0.94; CI 95%: 0.91–0.97) and fewer changes in medication dosage (OR = 0.97; CI 95%: 0.95–1.00), whereas participants taking medication for the respiratory system (OR = 3.21; CI 95%: 1.63–6.32) were more likely to report an increase in medication dose.

Our results show that 7% (n = 61/827) of the participants reported that the pandemic had moderately to highly affected their motivation to take their medications. Older (with each year increase in age) (OR = 0.97; CI 95%: 0.94–0.99) and tertiary-educated participants (OR = 0.45; CI95%: 0.26–0.89) were less likely to report an impact on their motivation to take their medications. On the other hand, participants taking medication for the respiratory system (OR = 2.12; CI95%: 1.04–4.32); those taking more than three long-term medications (OR = 2.55; CI95%: 1.28–5.05); and individuals feeling more at risk of COVID-19 because of their long-term condition (OR = 2.69; CI95%: 1.29–5.64) were more likely to report an effect on such motivation. The results of the goodness-of-fit test conducted in this analysis demonstrate statistical adequacy with the observed data, as indicated by a *p*-value exceeding 0.05.

**Table 3.** Multivariate analysis investigating the association between the impact of the COVID-19 pandemic on medication adherence and sociodemographic and clinical variables.

| | Medication Adherence OR (CI 95%) | | | |
|---|---|---|---|---|
| | **Average to Extreme Impact on Motivation to Take Medication (vs. Limited or No Impact) Sample Size: 708 HL Goodness of Fit (p-Val): 0.21** | **Change in Medication Intake Regularity (More or Less Regularly vs. No Change) Sample Size: 691 HL Goodness of Fit (p-Val): 0.95** | **Change in Medication Dosage (Increase or Decrease vs. No Change) Sample Size: 696 HL Goodness of Fit (p-Val): 0.82** | **Medication Interruption or Delay Sample Size: 703 HL Goodness of Fit (p-Val): 0.73** |
| **Female (vs. male)** | 0.96 (0.51–1.82) | 1.46 (0.76–2.80) | 1.74 (0.93–3.26) | 1.23 (0.62–2.44) |
| **Age (continuous)** | 0.97 (0.94–0.99) * | 0.96 (0.94–0.98) ** <br> More regular: 0.99 (0.96–1.01) <br> Less regular: 0.93 (0.89–0.96) ** <br> (Sample size = 795; HL Goodness of fit (p-val): 0.11) | 0.98 (0.95–1.00) * <br> Increase: 0.98 (0.96–1.01) <br> Decrease: 0.95 (0.92–0.98) ** <br> (Sample size: 797; HL Goodness of fit (p-val): 0.87) | 0.96 (0.94–0.99) * <br> Interruption: 0.94 (0.91–0.97) ** <br> Delay: 0.99 (0.96–1.02) <br> (Sample size: 811; HL Goodness of fit (p-val): 0.54) |
| **Tertiary education (vs. primary and secondary)** | 0.48 (0.26–0.89) ** | 0.61 (0.33–1.10) | 0.80 (0.46–1.39) | 1.51 (0.80–2.86) |
| **Living as a single adult (vs. living with at least one adult)** | 1.20 (0.59–2.44) | 1.13 (0.56–2.29) | 1.30 (0.68–2.47) | 1.22 (0.59–2.55) |
| **Poor to very poor general health (vs. average to very good)** | 1.30 (0.42–4.07) | 0.58 (0.12–2.85) | 1.65 (0.48–5.65) | 2.64 (0.82–8.50) <br> Interruption: 1.94 (0.42–8.99) <br> Delay: 3.54 (0.99–12.60) <br> (Sample size: 811; HL Goodness of fit (p-val): 0.54) |
| **Not vaccinated against COVID-19 (vs. 1 or 2 doses)** | 1.27 (0.61–2.67) | 0.50 (0.21–1.20) | 0.85 (0.40–1.82) | 1.56 (0.71–3.41) |
| **Perceived risk of COVID-19 due to chronic disease (vs. no perceived risk)** | 2.69 (1.29–5.64) * | 0.94 (0.37–2.38) | 1.03 (0.45–2.37) | 1.32 (0.56–3.16) |
| **Taking > 3 chronic medications (vs. 3 or less)** | 2.55 (1.28–5.05) ** | 1.76 (0.87–3.58) | 0.89 (0.42–1.89) | 1.51 (0.70–3.28) |
| **Medication for the respiratory system (vs. all other medications)** | 2.12 (1.04–4.32) * | 2.92 (1.49–5.72) ** <br> More regular: 4.27 (2.11–8.63) ** <br> Less regular: 1.09 (0.34–3.47) <br> (Sample size = 795; HL Goodness of fit (p-val): 0.11) | 2.33 (1.20–4.55) * <br> Increase: 3.21 (1.63–6.32) ** <br> Decrease: 0.72 (0.21–2.55) <br> (Sample size: 797; HL Goodness of fit (p-val): 0.87) | 0.67 (0.26–1.72) |

* $p < 0.05$ ** $p < 0.01$.

*3.2. Impact of the Pandemic on Access to Medications, Medical Appointments, Exams and Overall Medical Care*

Among the participants, 64 (8%) reported stockpiling their long-term medication for a period of three months or longer, while 9 (1%) temporarily had no access to their medication. Each year increase in age decreases the odds of stockpiling medication for more than 3 months (OR = 0.97; CI 95%: 0.94–0.99), whereas taking more than three medications (OR = 2.92: CI 95%: 1.50–5.64) and feeling more at risk of COVID-19 infection because of a long-term condition (OR = 2.38; CI 95%: 1.13–5.02) increased such odds, as shown in Table 4. Seventeen percent (n = 179/1054) of reported medical appointments were either postponed (n = 87), conducted less frequently (n = 81) or canceled (n = 11). Among the main reasons given for less frequent appointments were fear of being infected (36%) and currently being infected with COVID-19 (14%). Additionally, 14% of the appointments were conducted remotely, and 8% of the participants reported having had fewer medical exams because of the pandemic. Participants with tertiary education (OR = 1.91; CI 95%: 1.23–2.96), those who rated their general health as poor (OR = 4.10; CI 95%: 1.60–10.46) and those taking more than three long-term medications (OR = 1.97; CI 95%: 1.16–3.35) were more likely to report a negative impact on their medical appointments (postponed, canceled or conducted less frequently), while older participants (with each year increase in age) were less likely to report such a negative impact (OR = 0.96 CI 95%: 0.95–0.98). The results of the goodness-of-fit test conducted in this analysis demonstrate statistical adequacy with the observed data, as indicated by a *p*-value exceeding 0.05.

**Table 4.** Multivariate analysis to investigate the association between sociodemographic and clinical variables and the impact of the pandemic on overall medical care, access to physician appointments and medication.

| | Overall Medical Care and Access to Physicians and Medication OR (CI 95%) | | | |
|---|---|---|---|---|
| | **Strong to Extreme Impact on Overall Medical Care** *(vs. Limited or No Impact)* Sample Size: 695 HL Goodness of Fit *(p-Val): 0.22* | **Average Impact on Overall Medical Care** *(vs. Limited or No Impact)* Sample Size: 695 HL Goodness of Fit *(p-Val): 0.22* | **Physician Appointments Postponed, Canceled or Conducted Less Frequently** *(vs. No Impact)* Sample Size: 665 HL Goodness of Fit *(p-Val): 0.14* | **Need for Stockpiling Medications for More Than 3 Months** *(vs. Always Accessing Medications)* Sample Size: 693 HL Goodness of Fit *(p-Val): 0.38* |
| **Female** *(vs. male)* | 0.82 (0.39–1.74) | 1.95 (1.11–3.42) * | 1.35 (0.86–2.12) | 1.50 (0.78–2.87) |
| **Age** *(continuous)* | 0.96 (0.93–0.99) ** | 0.97 (0.95–0.99) ** | 0.96 (0.95–0.98) ** | 0.97 (0.94–0.99) ** |
| **Tertiary education** *(vs. primary and secondary)* | 1.20 (0.59–2.47) | 0.66 (0.40–1.09) | 1.91 (1.23–2.96) ** | 0.96 (0.53–1.72) |
| **Living as only adult** *(vs. living with at least one adult)* | 3.73 (1.80–7.73) ** | 0.97 (0.52–1.80) | 1.46 (0.88–2.42) | 1.28 (0.65–2.54) |
| **Poor to very poor general health** *(vs. average to very good)* | 3.73 (1.11–12.52) * | 4.19 (1.55–11.35) ** | 4.10 (1.60–10.46) ** | 0.53 (0.13–2.07) |
| **Not vaccinated against COVID-19** *(vs. 1 or 2 doses)* | 0.99 (0.37–2.62) | 0.75 (0.37–1.51) | 0.90 (0.49–1.64) | 0.76 (0.33–1.74) |
| **Perceived risk of COVID-19 due to chronic disease** *(vs. no perceived risk)* | 2.81 (1.16–6.78) * | 1.38 (0.67–2.86) | 1.75 (0.96–3.19) | 2.38 (1.13–5.02) * |
| **Taking more than 3 chronic medications** *(vs. 3 or less)* | 3.41 (1.56–7.43) ** | 2.32 (1.29–4.15) ** | 1.97 (1.16–3.35) * | 2.92 (1.50–5.64) ** |
| **Medication for the respiratory system** *(vs. all other medications)* | 1.75 (0.71–4.33) | 1.73 (0.89–3.33) | 1.29 (0.71–2.33) | 1.27 (0.58-2.75) |

* *p* < 0.05 ** *p* < 0.01

Seventeen percent (n = 165/982) of participants reported that the pandemic had an impact (average to extreme) on their overall medical care. The odds of perceiving a strong to extreme impact on overall medical care were significantly reduced with each year of increasing age (OR = 0.96; CI 95%: 0.93–0.99). Conversely, living as a single adult (OR = 3.73; CI 95%: 1.80–7.73), poor general health (OR = 3.73; CI 95%: 1.11–12.52), feeling at risk of

COVID-19 because of a long-term condition (OR = 2.81; CI 95%: 1.16–6.78) and taking more than three medications (OR = 3.41; CI 95%: 1.56–7.43) significantly increased the odds of perceiving such an impact.

*3.3. Impact on Diet and Physical Activity Programs, Participation in Therapeutic Support Groups and Home Care*

Among people who received home care, 19% (n = 12/62) had their home care reduced or stopped because of the pandemic. Participation in therapeutic support groups and associations was impacted by the pandemic for 28% (n = 68/306) of participants. This impact was increased among participants who took more than three long-term medications (OR = 3.04; CI 95%: 1.50–6.16) (Table 5). Among participants who had to follow a recommended diet or physical activity, respectively, 24% (n = 135/557) and 46% (n = 286/628) reported an impact from the pandemic. Each year increase in age is significantly associated with lower odds of reporting an impact on both diet (OR = 0.95; CI 95%:0.93–0.97) and physical activity (OR = 0.97; CI 95%:0.97–0.99). The impact of the pandemic on diet was higher in participants taking more than three long-term medications (OR = 1.93; CI 95%: 1.03–3.60) and those who felt more at risk of COVID-19 because of their chronic condition (OR = 2.32; CI 95%: 1.99–4.47). The results of the goodness-of-fit test conducted in this analysis demonstrate statistical adequacy with the observed data, as indicated by a *p*-value exceeding 0.05.

**Table 5.** Multivariate analysis for investigating associations between sociodemographic and clinical variables and the impact of the pandemic on diet, physical activity and therapeutic-related activities.

| | Diet OR (CI 95%) (Average to Extreme Impact) Sample Size: 380 HL Goodness of Fit (*p*-Val): 0.42 | Physical Activity OR (CI 95%) (Average to Extreme Impact) Sample Size: 422 HL Goodness of Fit (*p*-Val): 0.94 | Participation in Therapeutic Education/Patient Association Activities OR (CI 95%) Sample Size: 205 HL Goodness of Fit (*p*-Val): 0.94 |
|---|---|---|---|
| **Female (vs. male)** | 1.04 (0.60–1.79) | 1.44 (0.95–2.18) | 1.31 (0.66–2.63) |
| **Age (continuous)** | 0.95 (0.93–0.97) ** | 0.97 (0.97–0.99) ** | 0.99 (0.97–1.02) |
| **Tertiary education (vs. primary and secondary)** | 1.31 (0.78–2.20) | 1.15 (0.76–1.73) | 0.89 (0.45–1.76) |
| **Living as only adult (vs. living with at least one adult)** | 0.90 (0.47–1.72) | 0.93 (0.56–1.52) | 1.06 (0.50–2.28) |
| **Perceiving financial difficulties** | 1.92 (0.91–4.06) | 1.48 (0.76–2.90) | 1.33 (0.45–3.95) |
| **Poor to very poor general health (vs. average to very good)** | 1.01 (0.30–3.43) | 3.24 (0.82–1.87) | 2.78 (0.74–10.48) |
| **Not vaccinated against COVID-19 (vs. 1 or 2 doses)** | 0.90 (0.42–1.91) | 0.72 (0.39–1.34) | 1.24 (0.47–3.27) |
| **Perceived risk of COVID-19 due to chronic disease (vs. no perceived risk)** | 2.32 (1.99–4.47) * | 1.64 (0.90–2.96) | 0.77 (0.28–2.17) |
| **Taking more than 3 chronic medications (vs. 3 or less)** | 1.93 (1.03–3.60) * | 1.17 (0.90–2.96) | 3.04 (1.50–6.16) ** |

\* *p* < 0.05 ** *p* < 0.01.

## 4. Discussion

Our study shows that the COVID-19 pandemic and lockdown measures had a limited impact on patients' adherence to their long-term medications, their motivation to take their medications and access to medications, as well as on their overall medical care in the Geneva area. Through the experiences of individuals with various long-term diseases, these results contribute to a better understanding of the potential challenges faced by the Swiss healthcare system in terms of patient access to care and medication adherence during a pandemic and could help to better prepare for future health crises.

Rather than focusing on a single aspect of medication adherence, such as initiation, implementation or discontinuation, or factors associated with adherence, such as access to medications or shortages, our study provides a broader vision of medication intake during the COVID-19 pandemic by evaluating several of these aspects distinctly. Moreover, our results also differentiate individuals who discussed any non-adherent behavior such as interruptions or dosage modifications with a healthcare professional from those who did not seek any advice.

The prevalence of patients who reported a change in their medication intake during the COVID-19 pandemic is significantly lower in our study when compared with studies conducted among long-term patients in the US [29,30] but similar to other studies conducted in Europe [31–36]. Older and highly educated individuals were less likely to see their medication intake impacted by the pandemic, which is consistent with other studies [29,37], while taking medication for the respiratory system increased, by over three times, both the odds of having more regular medication intake and of increasing the medication dosage. Moreover, in our study, participants taking medication for the respiratory system, those taking more than three long-term medications and those who perceived increased risks because of their long-term conditions were also more likely to report an impact on their motivation to take their medications. This can be linked to the individuals' concerns and perceived increased risks of developing severe COVID-19 symptoms because of a long-term disease, which may differ according to the type of disease and increase with the number of medications [38,39]. The literature describes several reasons for modifying or discontinuing treatment, including fear of being infected by attending hospitals or doctor appointments, a lack of access to medications and a perceived risk of severe COVID-19 symptoms due to the immunosuppressive effect of some medications [4,40]. In addition, our results show that more than 30% of medication discontinuations or adjustments were conducted without discussion with a healthcare professional, which is higher than reported in two other studies [29,31]; the literature in this area is lacking and needs further attention.

The COVID-19 pandemic presented challenges in terms of access to long-term medications, primarily because of lockdown measures, limited access to healthcare services and an increase in drug shortages [29,41,42]. In our study, 8% of participants stockpiled their long-term medications in quantity for more than three months—the maximum quantity authorized in Switzerland—while only 1% reported not being able to access their medications, which is lower than what was reported elsewhere [2,29,43]. This can be partly explained by the fact that people were not prevented from leaving their homes but rather strongly advised to stay home (a partial lockdown); this may be an interesting element to consider when preparing for future health crises [44]. Other known reasons for poor medication access were non-renewed prescriptions due to canceled appointments with a physician [4]. In Switzerland, access to long-term prescription medications has been guaranteed since January 2019 by community pharmacists who are authorized to renew and dispense long-term treatments without a prescription if the patient cannot access the prescriber for any reason [45]. Synergistically, pharmacists provide home deliveries of medications and conduct telephone interviews for specific adherence programs, while physicians issue a greater number of electronic prescriptions, as also reported in other studies [46–50].

The pandemic and lockdown measures had a negative impact on nearly 20% of medical appointments, resulting in postponed and, in some cases, canceled appointments. This number is high when compared with another study conducted in Geneva in November 2020, which showed that 8% of participants had forgone healthcare since the beginning of the pandemic [22] yet lower than other studies conducted, for example, in the US and China [51,52]. Individuals who rated their general health as poor and those taking more than three long-term medications were, respectively, 4.1 and 1.9 times more likely to report an impact on medical appointments with a potential risk of increasing inequality in access to healthcare. It is, therefore, essential to ensure access to care and resources for all patients, especially those in vulnerable populations [53]. Our participants reported that 14% of

medical appointments were conducted remotely; this proportion is not negligible. To the best of our knowledge, no other study has evaluated the use of virtual health among individuals with long-term diseases in Switzerland. According to a study conducted in a Swiss walk-in emergency care unit, no significant increase in virtual care was observed during or after the COVID-19 pandemic [54]. Such developments are likely to be underway and need to be further evaluated. In other countries, however, there was an increase in the utilization of virtual care (telemedicine/telehealth), driven by the need to reduce the risk of COVID-19 transmission through in-person interactions but also to provide access to care for patients unable to leave their homes [55,56]. Although more than 50% of the population in our study had a tertiary education, it is important to acknowledge that the potential of digital care may be limited for people with lower levels of education or digital literacy [57].

Unexpectedly, in our study, each year of age increase was associated with a lower perception of the impact of the pandemic on several outcomes. It is worth noting that only 2% of our sample was over 81 years old, and few individuals reported poor general health and issues with mobility and autonomy, which are typically indicators of frailty among older people [58]. In the literature, Pasion et al. also showed that older people perceived less impact of COVID-19 on their overall healthcare compared to middled aged adults. [59]. We can also hypothesize that the system installed to protect the elderly population partially contributed to decreasing the perceived impact of the pandemic, as research conducted in Switzerland has shown a good understanding of and adherence to the lockdown measures [60].

Interestingly, not being vaccinated has no impact on access and adherence to medication or medical appointments. At the time of the study, the COVID-19 vaccine was available for free to the entire population regardless of age or comorbidities, and 75% of participants had received at least one dose of the COVID-19 vaccine. However, it is important to point out that access to care is ensured regardless of an individual's vaccinal status. To enter healthcare settings such as medical centers, hospitals or community pharmacies, patients were required to wear face masks; maintain physical distance; use hand sanitizer; and, in some cases, provide a negative COVID-19 test [61].

Consistent with the literature, nearly 20% of participants benefiting from home care services reported a reduction or interruption in such help due to the pandemic and because of workforce shortages, which are associated with an increase in demand, fear of exposure and containment measures [62,63]. Additionally, our results show a significant impact on participation in therapeutic support groups and associations. During the lockdown period, patient associations were forced to suspend their activities, which increased isolation for patients with long-term diseases. Finally, our study shows the impact of the pandemic on recommended diet and physical activities, which is consistent with other studies reporting on self-care measures [64,65]. Lockdown measures such as stay-at-home orders and quarantine often limited opportunities for physical exercise and increased sedentary behavior, as well as changes in dietary patterns such as increases in overeating [66,67].

Rather than focusing on one specific disease or type of population [4], our questionnaire was designed to evaluate the impact of the COVID-19 pandemic on any type of individual with a long-term disease among the general population. The population-based sample and its size allowed us to collect a considerable amount of data, which helped to better analyze the impact of the pandemic on individuals' access to care and medications and medication adherence.

This study has some limitations. First, there is an overrepresentation of women and highly educated individuals among the participants in the study and an under-representation of individuals reporting financial difficulties and older participants (>75 years old), which are known selection biases in online surveys that are difficult to manage [68,69]. Second, self-report questionnaires induce a risk of overreporting adherence because of social desirability and memory bias [70]. This is counterbalanced by the fact that the questionnaire was filled online, at a time and in an environment chosen by the participant, which is dissociated from a medical visit, and participants were informed that data were pseudonymized. Third, the

survey was conducted from September to November 2021, with questions referring to the impact of the pandemic since its start in March 2020. At the time of the survey, several measures to prevent the spread of COVID-19 were still observed in Geneva, such as the mandatory use of face masks, COVID-19 certificates to access indoor public places and events and remote work whenever possible. Although a recall bias cannot be ruled out, the time between the survey and the start of the pandemic may have enabled participants to answer the questions with more distance, experience and confidence. Fourth, the question related to the impact of the pandemic on overall medical care, motivation to take medications, physical activity and diet investigated the impact without disentangling the positive from the negative nature of the impact, which may be subject to the participants' interpretation of the word "impact". Fifth, the main long-term diseases from which participants suffered are not fully aligned with our national statistics, which place malignant tumors, cardiovascular diseases and depression as, respectively, the first-, second- and third-highest long-term diseases among the Swiss population. This can be partly explained by a selection bias associated with online questionnaires, in which individuals suffering from specific diseases may not feel well enough to participate, which is illustrated in the very small percentage of participants reporting poor general health. Patients with cancers were underrepresented in our sample, as well as the elderly, even though they were invited to participate. This may result from a variation in the perception of cancers as middle-term diseases rather than long-term ones. This perception may vary based on the types and stages of cancer, as well as personal experiences with the disease, particularly stigma and other priorities in life [71,72].

## 5. Conclusions

The lessons learned from this study can inform future preparedness strategies for healthcare systems, emphasizing the need to ensure access to long-term medications and outpatient care during health crises. Our study shows that the pandemic had a limited perceived impact on medication adherence for patients with long-term diseases. Access to long-term medications has been ensured through measures such as partial lockdowns—which did not prevent individuals from leaving their homes—as well as changes in healthcare utilization—such as telehealth—and community pharmacist-led renewals of prescriptions. Our results also showed that some patients endorsed a non-adherent behavior by discontinuing or modifying the treatment intake without a healthcare professional's recommendation. This crucial aspect of medication adherence should be investigated further to target interventions that could help individuals with long-term conditions to better manage their medications. Communication and collaboration between patients and interprofessional healthcare providers regarding self-management are of prime importance during a pandemic.

**Supplementary Materials:** The following supporting information can be downloaded at https://www.mdpi.com/article/10.3390/covid4020015/s1: "Questionnaire Participants with Chronic Illnesses".

**Author Contributions:** B.S., M.P.S., H.B. and S.S. conceptualized and designed the medication adherence and access to care questionnaire. H.B., N.P., I.G. and S.S. prepared the data collection and provided access to the general Specchio-COVID-19 dataset. B.S., Y.B., S.G. and M.P.S. prepared the database and managed the data for the analysis. Y.B. and S.G. conducted the statistical analysis. B.S. and M.P.S. wrote the manuscript. All authors have read and agreed to the published version of the manuscript.

**Funding:** No funding was received to conduct this study. The Specchio-COVID-19 study is funded by the Swiss Federal Office of Public Health, the General Directorate of Health of the Department of Safety, Employment and Health of the canton of Geneva, the Private Foundation of the Geneva University Hospitals, the Swiss School of Public Health (Corona Immunitas Research Program), and the Fondation des Grangettes.

**Institutional Review Board Statement:** The Specchio-COVID-19 study was approved by the Cantonal Research Ethics Commission of Geneva, Switzerland (Project ID 2020-00881).

**Informed Consent Statement:** Informed consent was obtained from all subjects involved in the study.

**Data Availability Statement:** The data are available upon request from the corresponding authors. The ethics protocols under which the data were collected do not permit public data deposition.

**Acknowledgments:** The authors thank all the participants of the Specchio-COVID-19 and the Specchio-COVID-19 study group.

**Conflicts of Interest:** The authors declare no conflicts of interest.

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
