# Peer review of "Patient-Perceived Impact of the COVID-19 Pandemic on Medication Adherence and Access to Care for Long-Term Diseases: A Cross-Sectional Online Survey"

_covid, doi:10.3390/covid4020015_

Round 1

Reviewer 1 Report

Comments and Suggestions for Authors

11.      I have a concern regarding the title of this manuscript.  The study results should compare with those outcomes in the pre-covid period. However, this is a cross-sectional study with all study participants being under COVID Lockdown. The study results only revealed what had happened during the COVID period.   In all logistic regression models, I am not able to identify a variable or variables that would allow me to discuss the “Impact of the COVID -19 pandemi...” , instead, the results are more of patients’ experience with COVID pandemic and lockdown.  I even hope that I can locate a proxy variable or variables in those models to describe the levels of lockdown experience.

22.      In the method section, there should be a brief description of the rationales why those study variables were included in the logistic regression model. .    I suggest you create an interaction variable (age group by living as a result) to reflect the impact of lock down on overall medical care. The authors may have better suggestions based on the data set they possessed.

33.      Lines 203-204   “ while older participants(with each year increase in age) were less likely to report such a negative impact (OR=0.97 CI95%:0.95-0.99”   Please show the range of age  [56 (IQR 47-66)] in the demographic section.   Since there is a selection bias of on-line survey samples, the older participants may have different connotation in this manuscript.  I am not sure if they are seniors living as single adults or having mobility difficulties.

44      Lines 206-211  “Odds of perceiving a strong to extreme impact on overall medical care were significantly reduced with each year of increasing age (OR=0.96; CI95%: 0.93-0.99). Conversely, living as a single adult (OR=3.20; CI95%: 1.52-6.76), poor general health (OR=5.76; CI95%:1.76-18.81) and taking more than three medications (OR=3.10; CI95%:1.39-6.89) significantly increased the odds of perceiving such impact.” 

The authors should explain some unexpected results in this manuscript. A case in point is in Lines 206-211.   I assume increasing age may exert a significant impact on overall medical care. 

55.      Lines 194-204.  What are the units of analyses Here?  Are they appointment or patient?

66.      I am not sure why you have an education level---primary, which only accounts for 4% of study samples, as a group and even as a reference group in the model.  Since you have a selection bias in this study, I suggest that you just use two levels, tertiary vs secondary and primary.

77.      Please explain why perceived risk of the covid action due to own chronic disease is not included in all regression models.

88.      All of the model results tables should include Sample Size n=???.  I have challenges to determine who was included in each logistic model analysis.   I don’t think 982 participants were included in each model in all Tables. 

99.      Based on the structure of the Switzerland health care system, the pharmacist plays a pivotal role in access to care and medications.  How do you accurately reflect the pharmacist’s influence in the study model even you don’t have the data?  Also, please discuss the extent of use virtual care in Switzerland in 2021.  (Line 197)

110.  Please include goodness of fit of logistic regression models to see how well all models fit the study data.  

111.  Line 291  please explain why this increasing risk of health disparity occurred?

112.   I am surprised that perceived financial difficulties and vaccination status had no significant impact of in the model.    I am not sure I would agree this statement as suggested by the authors that the use of PPEs may be attributable to no vaccination impact.  

Author Response

Thank you very much for taking the time to review this manuscript. Please find the detailed responses as attached file and the corresponding revisions/corrections highlighted/in track changes in the re-submitted files (line number refers to track changes version)

Reviewer 2 Report

Comments and Suggestions for Authors

In the abstract add some percentages in addition to the odds ratio when possible to make the results more explanatory

In the Introduction elaborate with more emphasis on the importance and significance of the study

In the material and methods section add the questionnaire as supplementary data

More information about the numbers of the participants and on what basis the number was calculated is need to be explained in the material and method section. Moreover, the inclusion and exclusion criteria must be more explained in the material and methods section

In the results section one point that is not clear to me why respiratory medication particularly was used to be tested against other medication.  This point must be discussed and to be clearer for the reader

The conclusion part should be expanded to include son of the significant results

Over all the manuscript has to be rechecked again for any grammatical and typing mistakes

The reference must be reached again to consistent and have the same format for example

Comments on the Quality of English Language

Minor corrections

Author Response

Thank you very much for taking the time to review this manuscript. Please see the attachment for the detailed responses and the corresponding revisions/corrections highlighted/in track changes in the re-submitted files.

Reviewer 3 Report

Comments and Suggestions for Authors

The paper „Impact of the Covid-19 pandemic on medication adherence and access to care in patients with long-term diseases: A cross sectional online survey“, assesses the impact of the COVID-19 pandemic on patients with various long-term diseases, rather than focusing on patients with one specific disease. Risking a selection bias associated with online questionnaires, in which patients suffering from specific diseases may be unwilling to participate, this study presents comprehensive and overall assessment of the impact of the pandemic on patients’ medication intake, access to medicines, medical appointments and home care services. The results obtained in this study can potentially contribute to a better understanding of patient access to care and medication adherence during a pandemic.

Author Response

Thank you very much for taking the time to review this manuscript. Please see the attachment for the detailed response.

Round 2

Reviewer 1 Report

None

None